# Detection and Morphological Analysis of Micro-Ruptured Cortical Arteries in Subdural Hematoma: Three-Dimensional Visualization Using the Tissue-Clearing Clear, Unobstructed, Brain/Body Imaging Cocktails and Computational Analysis Method

**DOI:** 10.3390/diagnostics12112875

**Published:** 2022-11-20

**Authors:** Kazuhisa Funayama, Kazuki Tainaka, Akihide Koyama, Rieka Katsuragi-Go, Natsumi Nishikawa-Harada, Ryoko Higuchi, Takashi Aoyama, Hiraku Watanabe, Naoya Takahashi, Hisakazu Takatsuka

**Affiliations:** 1Division of Legal Medicine, Department of Community Preventive Medicine, Graduate School of Medicine and Dental Sciences, Niigata University, Niigata 951-8510, Japan; 2Center of Cause of Death Investigation, Graduate School of Medical and Dental Sciences, Niigata University, Niigata 951-8510, Japan; 3Department of System Pathology for Neurological Disorders, Basic Neuroscience Branch, Brain Research Institute, Niigata University, Niigata 951-8510, Japan; 4Department of Radiological Technology, Graduate School of Health Sciences, Niigata University, Niigata 951-8510, Japan

**Keywords:** subdural hematoma, cortical artery, arterial rupture, 3D imaging, CUBIC

## Abstract

One of the causes of bleeding in subdural hematoma is cortical artery rupture, which is difficult to detect at autopsy. Therefore, reports of autopsy cases with this condition are limited and hence, the pathogenesis of subdural hematoma remains unclear. Herein, for the detection and morphological analysis of cortical artery ruptures as the bleeding sources of subdural hematoma, we used the tissue-clearing CUBIC (clear, unobstructed, brain/body imaging cocktails and computational analysis) method with light-sheet fluorescence microscopy and reconstructed the two-dimensional and three-dimensional images. Using the CUBIC method, we could clearly visualize and detect cortical artery ruptures that were missed by conventional methods. Indeed, the CUBIC method enables three-dimensional morphological analysis of cortical arteries including the ruptured area, and the creation of cross-sectional two-dimensional images in any direction, which are similar to histopathological images. This highlights the effectiveness of the CUBIC method for subdural hematoma analysis.

## 1. Introduction

Cerebral contusion or bridging vein rupture resulting from severe head trauma is the main source of bleeding in acute subdural hematoma (SDH) [1,2,3]. Even minor subclinical trauma can cause SDH, which is called “nontraumatic” or “spontaneous” SDH [4,5,6]. One of the sources of bleeding in this type of SDH is the rupture of a cortical artery. Although 198 cases of SDH due to ruptured cortical arteries have been reported in 49 publications [3,5,7,8,9,10,11,12,13,14,15,16,17,18,19,20,21,22,23,24,25,26,27,28,29,30,31,32,33,34,35,36,37,38,39,40,41,42,43,44,45,46,47,48,49,50,51,52,53], only eight cases in two publications were subjected to detailed histopathological examination by autopsy [8,52]. Therefore, the pathogenesis of this condition remains to be conclusively established based on pathological findings.

Previous reports have indicated that most cortical artery ruptures were macroscopically missed at autopsy [14,52]; hence, histological examination of the rupture sites could not be performed. We previously reported that post-mortem computed tomography (CT) angiography and intra-arterial fluid perfusion of the isolated brain are useful in detecting arterial rupture at autopsy when it is the source of hemorrhage in SDH [52]. However, the arterial rupture was small, and its morphology was not visible macroscopically. Furthermore, the histological detection of arterial rupture requires hundreds or thousands of serial sections. These observations indicate that it is difficult to detect and investigate cortical arterial rupture using conventional macroscopic and microscopic methods at autopsy and that new methods are necessary to elucidate the pathogenesis of this condition.

Many researchers have developed various tissue-clearing methods since the high-speed three-dimensional (3D) imaging of cleared mouse brains by light-sheet fluorescence microscopy (LSFM) was first reported in 2007 [54]. Emerging tissue-clearing technologies have enabled comprehensive and high-resolution 3D visualization of biological tissue samples along with state-of-the-art microscopes [55,56], each individual method providing its own set of advantages and limitations.

The clear, unobstructed brain/body imaging cocktails and computational analysis (CUBIC) method combined with LSFM is a water-based tissue-clearing and imaging technique [57,58]. It is a relatively simple method in obtaining high-resolution 3D images of large human tissues, including the microvascular network [59,60]. The CUBIC method uses low-toxicity, water-soluble reagents for tissue-clearing, making it safe and less tissue-damaging than other techniques, and allows for post hoc histopathological analysis [61].

We aimed to determine whether the CUBIC method with LSFM is suitable for the 3D imaging of cortical arteries in the human brain and morphological analysis of cortical artery ruptures. In this study, we evaluated the usefulness and limitations of the CUBIC method for the detection and morphological analysis of ruptured cortical arteries. Comparing this method with conventional macroscopic and histological examination, we found that the CUBIC method has advantages over conventional methods in the investigation of arterial rupture as a source of bleeding in SDH.

## 2. Materials and Methods

This study was approved by the Research Ethics Committee of Niigata University (No. 2018-0224). The autopsies in this report were performed in accordance with the proper procedures under Japanese law, and all procedures were performed in accordance with the Declaration of Helsinki Principles. Since this study required past forensic autopsy records, and yet it was not possible to obtain informed consent for the use of the records from the subjects’ legal guardians, the Ethics Committee waived the need for written informed consent in this study. This decision was made in accordance with the “Ethical Guidelines for Medical Research Involving Human Subjects (enacted by the Ministry of Education, Culture, Sports, Science and Technology in Japan; and the Ministry of Health, Labor and Welfare in Japan),” Chapter 5, Section 12-1 (2) (a) and (c).

### 2.1. Human Brain Tissue

Among the forensic autopsy cases performed in our institution from 2017 to 2019, there were six cases of SDH (four acute, one subacute and one encapsulated acute (acute on chronic) cases) with cortical artery rupture detected histopathologically. All of these cases were included in this study (64–95 years old, average 86.7 years, one man and five women).

### 2.2. Macroscopic Imaging

Based on our previous report [52], we identified a suspected cortical artery rupture region with postmortem CT angiography and intra-arterial fluid perfusion during autopsy. The whole brain removed at autopsy was formalin-fixed and then cut into a rectangular block (26–55 mm × 18–50 mm × 12–26 mm) to include the region. Before and after brain tissue sampling, the suspected region was photographed close-up with a digital camera (D810 and AF-S Micro NIKKOR 60 mm, Nikon, Tokyo, Japan) to obtain magnified macroscopic images.

### 2.3. CUBIC Tissue-Clearing Reagents

To clear the brain block samples containing cortical arteries using the CUBIC method, CUBIC reagents were prepared according to previous reports [62,63]. The reagent CUBIC-L (T3740, Tokyo Chemical Industry, Tokyo, Japan), used for decoloring and delipidation, comprises 10 wt% N-butyldiethanolamine and 10 wt% Triton X-100 in distilled water. The reagent CUBIC-R (Tokyo Chemical Industry), used for clearing, comprises 45 wt% antipyrine (D1876, Tokyo Chemical Industry) and 30 wt% nicotinamide (N0078, Tokyo Chemical Industry). The pH 8–9 was adjusted by N-butyldiethanolamine (B0725, Tokyo Chemical Industry) in distilled water.

### 2.4. CUBIC Clearing of Brains

The block of tissue was made transparent using the CUBIC method (Figure 1). Each fixed brain block was washed with a sufficient volume of phosphate-buffered saline (PBS) to wash out the formalin for 6 h before clearing. Then, for decoloring and delipidation, each block was immersed in CUBIC-L at 45 °C with gentle shaking for 12 days. During this time, the CUBIC-L was replaced once. Each treated sample was then washed with PBS several times at room temperature. Consequently, the samples were immersed in CUBIC-R for 3 days for refractive index (RI) matching before LSFM imaging. After LSFM imaging, each sample was washed with PBS again and paraffin-embedded for histopathological examination.

### 2.5. LSFM Imaging Conditions

Each sample with RI matching was immersed in an oil mixture (RI = 1.525), as previously described [62], and imaged using LSFM (MVX10-LS, Olympus, Tokyo, Japan). A 0.63× objective lens (numerical aperture = 0.15, working distance = 87 mm) with 532 nm and 637 nm emission lasers was used to acquire the images, which were scanned in 5–10-μm stacks.

### 2.6. Serial Sectioning and Histopathological Examination

After LSFM imaging, the tissue was made detranslucent by PBS washing, and serial sections were prepared with Elastica van Gieson staining. Histopathological images were obtained under a light microscope (ECLIPSE Ci-L, Nikon) to detect arterial rupture.

### 2.7. Image Analysis

Image analysis was performed using a computer equipped with Imaris software (version 9.5.1, Bitplane, Zürich, Switzerland). From the image data obtained by LSFM, we created 3D and cross-sectional images of arterial rupture regions identified by histopathological examination and compared them with magnified macroscopic and histopathological images.

## 3. Results

In cases 1–5, histopathology confirmed that arterial ruptures had occurred at or near the root of branches that originate from a cortical artery near the Sylvian fissure. Using the CUBIC method, we were able to visualize the arterial rupture clearly in four cases. In case 5, the rupture was out of CUBIC imaging range. In case 6, histopathology confirmed a small arterial tear in the focal contusion contained a hematoma, which interfered with LSFM imaging of the tore artery. The results are summarized in Table 1.

### 3.1. Results Per Case

In cases 1 and 2, magnified macroscopic observation disclosed a defect in the cortical arterial wall with a small arterial branch, which was visualized in more detail in the CUBIC 3D images. In case 1, the CUBIC cross-sectional imaging reconstructed the morphology of the ruptured area with the same level of detail as the histopathological image (torn branches extending from the trunk of the cortical artery) (Figure 2).

In case 2, CUBIC cross-sectional images reconstructed the morphology of the ruptured cortical artery trunk as well as the histopathological image but indistinctly demonstrated the torn branch that was evident in the histopathological image (Figure 3).

In case 3, histopathology revealed two ruptures in one cortical artery: two thin arteries branching off from the cortical artery, one ruptured at its root and the other ruptured immediately after further branching. However, magnified macroscopic observations could not reveal the rupture or even the two small arterial branches due to the agglutination of bleeding on the surface. Clearly, 3D-CUBIC imaging showed the presence of two arterial branches near the cortical artery that lacked continuity with the cortical arterial trunk: one ruptured at its root and the other ruptured immediately after further branching. Anatomically, each rupture forms two edges of the ruptured artery: one on the proximal (cortical artery trunk) side and the other on the distal (small artery branch) side. Therefore, this case had four edges of the ruptured artery, all of which were detected in separate histopathological sections. The CUBIC images not only showed the same ruptures as the histopathological images, but also integrated all the rupture end sections into one image, which enables the observer to understand the interrelationship of the rupture sites (Figure 4).

In case 4, magnified macroscopic observation did not reveal the rupture of the cortical artery or small branch, despite bleeding deposits not being detected at the rupture site. Clearly, 3D-CUBIC imaging showed the cortical artery ruptured at the root of a small branch. The CUBIC cross-sectional image showed not only the ruptured cortical artery at the root of the small branch but also the arachnoid tear and the small nodule near the rupture as it appears in the histopathological section (Figure 5).

In case 5, arterial rupture was histologically detected outside the imaging range of CUBIC, so rupture findings could not be compared with CUBIC images.

In case 6, macroscopic observation revealed a focal contusion in the cerebral cortex of the left temporal lobe near the Sylvian fissure, and histopathology confirmed that the contusion contained a subcortical hematoma with a small arterial tear (Figure 6). CUBIC imaging showed the contusion of the cerebral cortex containing the hematoma but not the tear of the small artery. Unlike the other five cases in which the cortical artery had ruptured at the small branching bifurcation, in this case, the histopathological findings of the dura mater showed that a bridging artery connecting the dura to the brain was torn, causing the hemorrhage.

### 3.2. Implications of CUBIC Processing on Histological Examination

Overall, CUBIC processing had minimal effect on the histopathological observation of the artery in all cases, but small cracks occurred inside the brain parenchyma in some cases (Figure 7).

## 4. Discussion

Three-dimensional imaging using a tissue-clearing technique has been studied mainly in mouse organs and tissues [58,59]. However, there are also some studies on human tissues, and their interests include structure, distribution and network analysis in the nervous system [59,64,65,66,67]. In addition, they include cancer detection or structure analysis [61,67] and normal morphological analysis of the eye, thyroid and kidney [68], or the endometrium [69]. In this study of human brain tissue, we compared the combination of conventional gross and histopathological examinations with a tissue-clearing technique called CUBIC for the detection and morphological analysis of arterial rupture as a source of bleeding in SDH. This approach has not been attempted previously for SDH. We found that the CUBIC method can be used in most cases to visualize arterial structure with the same level of detail as histopathological examination and that the addition of the CUBIC method to the conventional methods provides several advantages.

Firstly, CUBIC analysis is better at detecting arterial rupture than conventional macroscopic observation. In addition to cases where the presence of a rupture was macroscopically apparent, such as case 2, CUBIC analysis allows for the clear visualization of ruptures even in cases where there were only small superficial changes that were difficult to recognize as a rupture. For instance, cases 1 and 4 where there was no evidence of a rupture at all, and in case 3, where the artery itself was hidden by surface bleeding. Particularly in case 3, the presence of two ruptures in one cortical artery could not be recognized macroscopically but could be clearly visualized using the CUBIC method. This means that even if one rupture is detected by conventional histological examination, there may be other ruptures that could be missed. The CUBIC method is not only capable of detecting multiple ruptures in a single image, but it also improves the detection sensitivity of arterial ruptures compared with conventional macroscopic observation [14,52].

Another advantage of the CUBIC method is that it allows for detailed 3D morphological analysis of arterial ruptures and their adjacent arteries. Even in cases where morphological changes in the ruptured arteries were unclear or absent in conventional macroscopic observation, the CUBIC method allowed us to visualize them with high-definition in 3D images and enabled detailed morphological analysis. Likewise, the CUBIC method enabled visualisation where the small branches around the ruptured area were too small or hidden by hemorrhage to be recognized. Since conventional histopathological examination is based on the morphology of thin tissue cross-sections, each section only holds 2D morphological information. Although it is possible to obtain morphological information from thick tissue blocks by preparing a large number of serial sections, it is not easy to integrate them to obtain high-definition 3D images, which the CUBIC method can create. In five of our six cases, rupture of the cortical artery occurred at the root of its small branch. The characteristic morphology of the arterial rupture, termed “fire hose” rupture [8], was presumed to be associated with the rupture occurrence due to potential anatomical weakness in the arterial bifurcation [13]. However, the specific pathogenesis is still unclear. Using CUBIC, we were able to perform 3D morphological analysis of arterial structure, which is difficult by conventional methods. This is expected to reveal the relationship of arterial structure to arterial rupture. In addition, from the CUBIC 3D image data it is possible to obtain any number of cross-sectional 2D images (and in any direction) [69]. In other words, the CUBIC method enables us to optimize the position and slice direction of pathological sections for morphological analysis by creating virtual tissue sections without destroying the tissue blocks. This feature has enabled efficient and effective pathological searches focusing on lesion sites. Moreover, this advantage is not limited to the specific condition of cortical artery rupture. It could be applied to other cerebrovascular disorders such as cerebral aneurysms or vertebral artery dissection, as well as to coronary artery disease, one of the most important conditions in forensic pathology.

However, the CUBIC method has some limitations that can occur not only under the specific condition of cortical artery rupture but also under common pathological conditions. The first limitation is that hematomas will not be sufficiently cleared, so their internal and surrounding structures will be obscured for LSFM imaging. Thin hemorrhages on the surface of the brain, as in case 3, did not interfere with the imaging of blood vessels, but in case 6, a subcortical hematoma around the ruptured artery inhibited the imaging of the lesion. The SDH itself can be removed from the brain during autopsy and does not interfere with CUBIC imaging. It would be difficult to adapt CUBIC imaging to conditions such as intracerebral hemorrhage or massive subarachnoid hemorrhage, where it is difficult to remove the hematoma during autopsy.

The second limitation is the vulnerability of the transparent tissue blocks. Tissue blocks become soft like jelly during the transparency process, so they must be handled carefully to avoid breakage. In our study, even careful handling of normal brain tissue resulted in the partial destruction of brain parenchyma, so the application of CUBIC imaging to more fragile tissue, such as brain tumors with necrosis, seems to be a destructive analysis which opposes the benefits of this method. Since arterial tissue is tougher than the brain parenchyma, none of the arteries were seriously affected in our case series. However, mishandling can break weak arteries.

Another limitation is that the size of tissue that can be imaged with CUBIC is limited by the condition of the tissue. Given that the fluorescence laser is scattered and attenuated as it travels through the tissue [70], the image will be obscured, especially in areas of insufficient transparency, such as bleeding sites. Therefore, if only a narrow area can be imaged with LSFM due to insufficient transparency, it is possible that the lesion is out of the imaging area in a single imaging session, as in case 5. However, in general, it is possible to obtain an entire tissue block of imaging data by repeating the narrow range of imaging in different positions and fusing the obtained image data computationally. Furthermore, the arterial rupture site can be detected by confirming extravascular leakage with post-mortem angiography CT or intra-arterial fluid injection, as we have reported previously [52]. In this technique, fluid (contrast medium) injection is manually performed while adjusting the injection pressure through a catheter inserted into the internal carotid artery, eliminating the need to directly access the ruptured cortical artery, and preventing an excessive increase in intravascular pressure from rapid fluid injection [52]. Hence, this technique ensures that the lesion is contained within a narrow imaging area of LSFM without causing artificial cortical arterial rupture.

Although the CUBIC method can create virtual tissue sections computationally, it cannot completely replace the histopathological examination in all cases. As is evident in case 2, cross-sectional images of the CUBIC method might not reveal fine structures. In addition, although immunostaining is possible with the CUBIC method [57,58], histopathological diagnosis typically requires various staining methods for the analysis of pathogenesis.

Based on the advantages and disadvantages described so far, we further discuss optics, statics, and analytical chemistry in our CUBIC-based analysis as below. Since forensic autopsy samples include coagulated blood, it is of primary importance to decolor the blood in order to execute clearing-based 3D imaging. The CUBIC cocktails consist of the chemical reagents optimized for eluting heme molecules from the coagulated blood [58,62]. Thus, CUBIC protocol provides broader optical windows available for imaging among the existing clearing protocols. On the other hand, the refractive index of CUBIC-R (RI = 1.52) is lower than those of DISCO-based clearing solvents (RI~1.56). The RI of tissue constituents varies over a range of 1.3 to 1.6 [71]. Therefore, the CUBIC-R may be difficult to eliminate the light scattering of hematoma assumed to have a high RI value.

Cleared samples show no apparent change in size over a year, even though the CUBIC-R slightly swells the tissue sample in the process of tissue-clearing [72]. Since CUBIC-L is a relatively strong base (pH~11), long-term 45 °C treatment may cause tissue fragility, such as the partial destruction of brain parenchyma observed in this study. On the other hand, if delipidation by CUBIC-L is insufficient, clearing by CUBIC-R will be poor. If possible, it is desirable to verify the vulnerability of samples due to the processing time of CUBIC-L and the sufficiency of transparency by CUBIC-R using test samples according to the sample conditions.

In analytical chemistry, we performed tissue autofluorescence imaging with excitation wavelengths in the visible region, which is effective for analytical chemistry such as chemical labelling and immunostaining. The autofluorescence of human brain tissue is extended over a broad wavelength band. Since the intensity of autofluorescence and the light transmittance after clearing highly depend on the sample condition, an arbitrary excitation wavelength was selected in our experiment. If the sample would be stained by specific labelling, the autofluorescence must be suppressed by hydrogen peroxide treatment prior to labelling, since autofluorescence causes background noise, but this process may enhance sample vulnerability.

The advantages and disadvantages of the CUBIC method described so far have only been observed in a few cases, and new advantages and disadvantages may become apparent when more cases are examined. This study does not describe the understanding of the pathogenesis itself of this type of SDH using the CUBIC method, and it is not clear at this time how much the CUBIC method can contribute to the elucidation of the pathogenesis. Based on our findings, we are currently applying the improved CUBIC method [62] to new cases as well as using conventional methods to elucidate the pathogenesis of this type of SDH.

## 5. Conclusions

In the investigation of arterial rupture as a source of bleeding in SDH, the CUBIC method clearly showed arterial rupture in 2D and 3D, although there were exceptions. The CUBIC method is a non-destructive method of tissue examination that does not hinder additional histopathological examination. This means that the addition of the CUBIC method to the conventional macroscopic observation and histopathological examination makes it possible to analyze the morphology of arterial rupture in more detail, which is useful for the analysis of SDH pathogenesis.

## Figures and Tables

**Figure 1 diagnostics-12-02875-f001:**
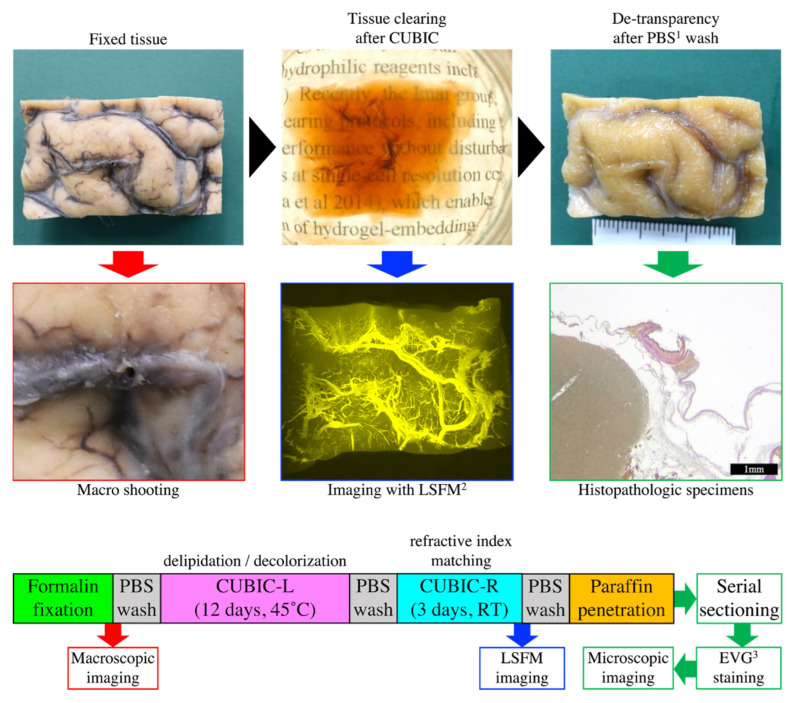
The procedural flow from tissue-clearing to histopathological analysis using the CUBIC method. ^1^ Phosphate-buffered saline, ^2^ Light-sheet fluorescence microscopy and ^3^ Elastica van Gieson.

**Figure 2 diagnostics-12-02875-f002:**
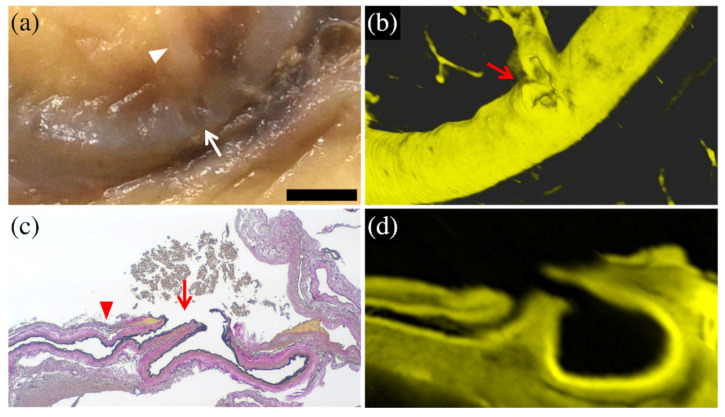
Comparison of macroscopic and histopathological images with CUBIC images in case 1. The enlarged macroscopic (formalin-fixed) image (**a**) showed the cortical artery with a small depression (arrow) and fine arterial branch (arrowhead). The 3D-CUBIC image (**b**) clearly showed the cortical artery ruptured at the base of a fine branch (arrow). The histopathological (Elastica van Gieson staining) image (**c**) showed a tear of the arterial wall (arrow) at the base of a fine arterial branch (arrowhead). The cross-sectional CUBIC image (**d**) reconstructed the morphology of the rupture as well as the histopathological image. The bar (on the lower right side of (**a**)) is equal to 2 mm for (**a**) and (**b**), 500 μm for (**c**), and 700 μm for (**d**). The panels (**a**,**c**) are modified from the study by Funayama et al. [52].

**Figure 3 diagnostics-12-02875-f003:**
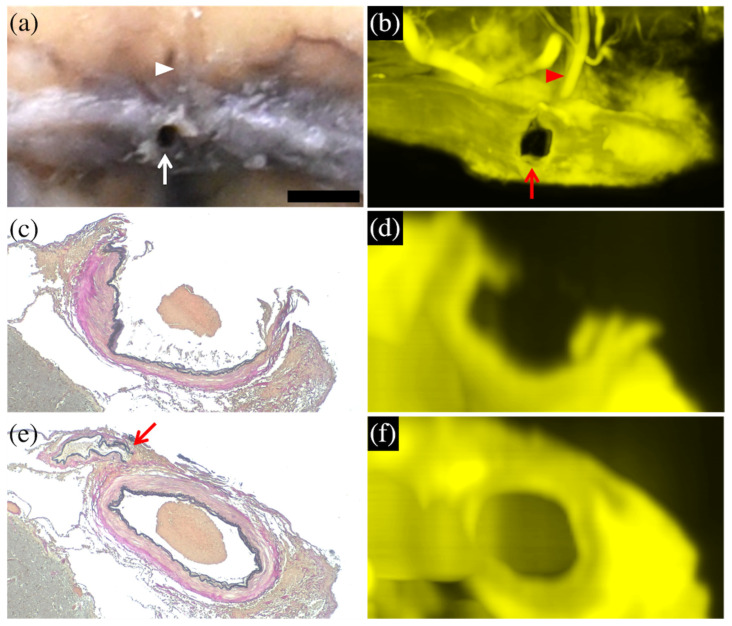
Comparison of macroscopic and histopathological images with CUBIC images in case 2. The enlarged macroscopic (formalin-fixed) image (**a**) shows the cortical artery with a small hole (arrow) and fine arterial branch (arrowhead), which was reconstructed and visualized in the 3D-CUBIC image (**b**). The histopathological (Elastica van Gieson staining) image (**c**) shows a wall defect in the cortical artery, which was reconstructed and visualized in the cross-sectional CUBIC image (**d**). Another histopathological (Elastica van Gieson staining) image (**e**) detected a torn end of a fine arterial branch (arrow), but this torn end is unclear in the cross-sectional CUBIC image (**f**). The bar (on the lower right side of (**a**)) is equal to 1 mm for (**a**,**b**) and 500 μm for (**c**–**f**).

**Figure 4 diagnostics-12-02875-f004:**
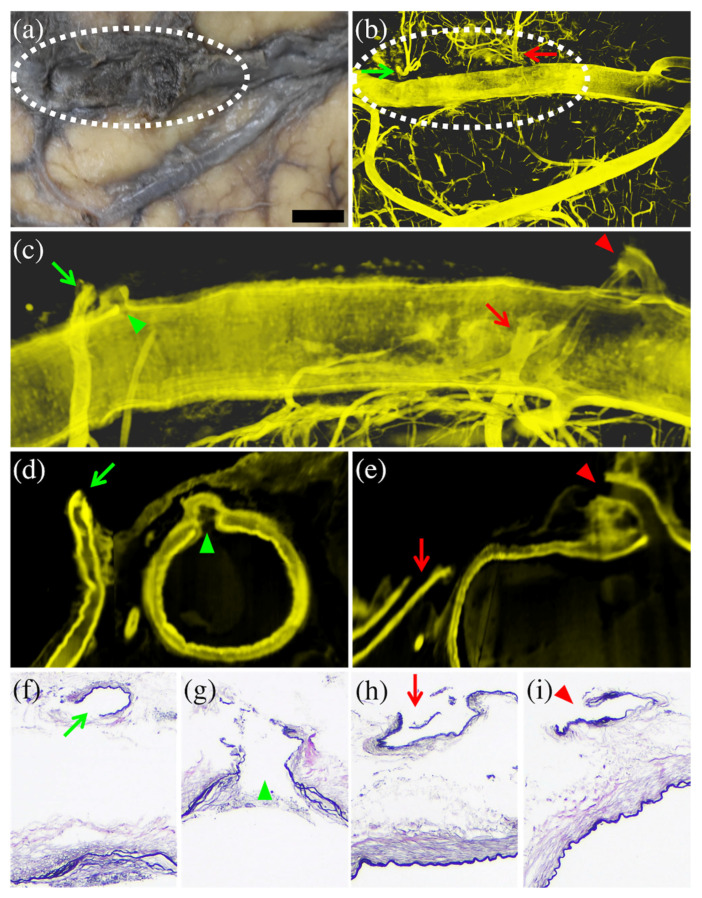
Comparison of macroscopic and histopathological images with CUBIC images in case 3. The enlarged macroscopic (formalin-fixed) image (**a**) could not reveal the presence of arterial ruptures or “twigs” due to bleeding on the surface (dotted circle). The 3D-CUBIC image (**b**) detected two small branches (arrows) at the bleeding site (dotted circle), neither of which was continuous with the cortical artery. The magnified lateral view of the 3D-CUBIC image (**c**) and the cross-sectional CUBIC images (**d**,**e**) of the bleeding area showed the proximal (arrowheads) and distal (arrows) ruptured ends of the two different arterial branches. These images indicated that one rupture occurred at the bifurcation of the cortical artery (green) and the other approximately 7 mm from the bifurcation (red) immediately after further branching. The histopathological (Elastica van Gieson staining) images (**f**–**i**) detected the four corresponding ruptures shown in the CUBIC images, but they could not display their positional relationship to each other. Each marker (arrow or arrowhead, green or red) in Figures (**b**–**i**) corresponds to each rupture part. The bar (on the lower right side of (**a**)) is equal to 2 mm for (**a**,**b**), 500 μm for (**c**), 400 μm for (**d**,**e**) and 200 μm for (**f**–**i**).

**Figure 5 diagnostics-12-02875-f005:**
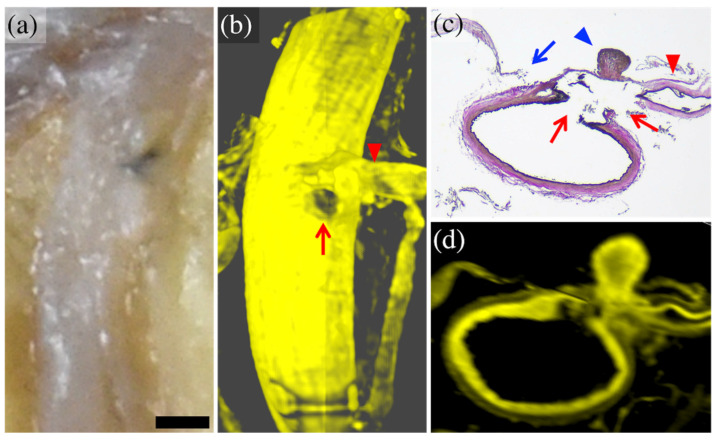
Comparison of macroscopic and histopathological images with CUBIC images in case 4. The enlarged macroscopic (formalin-fixed) image (**a**) did not reveal the presence of arterial rupture or a small arterial branch. The 3D-CUBIC image (**b**) clearly showed an arterial wall defect (arrow) at the base of a fine branch (arrowhead). The histopathological (Elastica van Gieson staining) image (**c**) showed the arterial rupture (red arrows) at the base of a fine arterial branch (red arrowhead) as well as a tear of the arachnoid (blue arrow) and a small nodule (blue arrowhead) near the rupture. The cross-sectional CUBIC image (**d**) reconstructed the morphology of the rupture, arachnoid tear and small nodule as shown by the histopathological image. The bar (on the lower right side of (**a**)) is equal to 0.5 mm for (**a**,**b**) and 300 μm for (**c**,**d**). The panel (**a**) is modified from the study by Funayama et al. [52].

**Figure 6 diagnostics-12-02875-f006:**
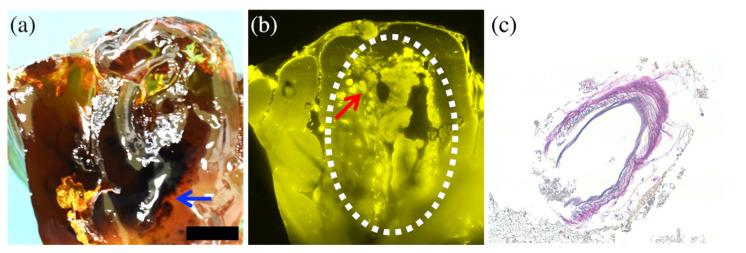
Comparison of macroscopic and the cross-sectional CUBIC image with histopathological image in case 6. The macroscopic image after CUBIC processing (**a**) showed an internal black hemorrhage (arrow) through the transparent brain parenchyma. The cross-sectional CUBIC image (**b**) showed the contusion of the cerebral cortex containing the subcortical hematoma (dotted circle). An inadequately clarified hematoma interfered with the CUBIC imaging of the ruptured artery, but there was a small torn artery at the margins of the hematoma ((**b**), arrow), which was detected in the histopathological (Elastica van Gieson staining) image (**c**). The bar (on the lower right side of (**a**)) is equal to 10 mm for (**a**,**b**) and 100 μm for (**c**).

**Figure 7 diagnostics-12-02875-f007:**
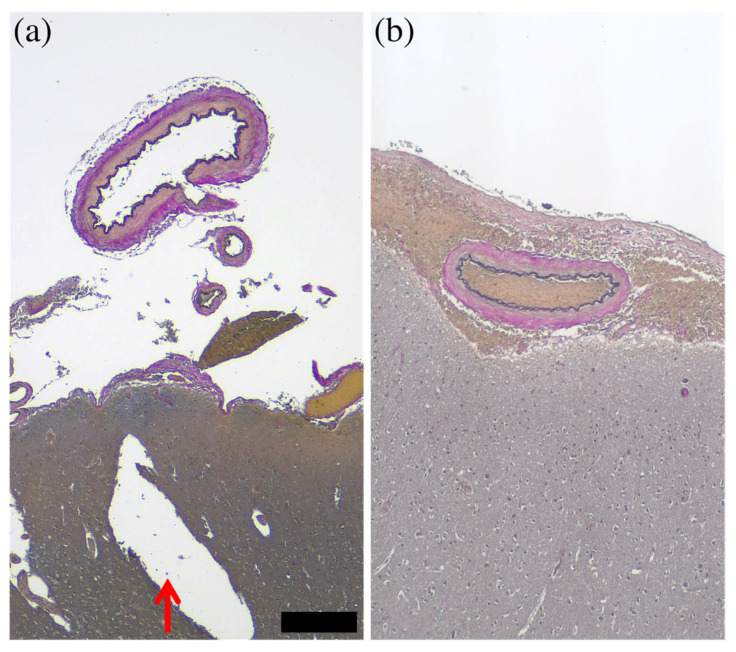
Comparison of histopathological images with/without CUBIC processing. Between the histopathological (Elastica van Gieson staining) images with (**a**) and without (**b**) CUBIC processing, there was little effect of CUBIC processing on the arteries, but the brain parenchyma was cracked (red arrow). This cracking was likely due to the weakening caused by CUBIC processing. The bar (on the lower right side of (**a**)) is equal to 300 μm for (**a**,**b**).

**Table 1 diagnostics-12-02875-t001:** Summary of the results.

No.	Age (Years), Sex	Type of SDH ^1^	Site of Arterial Rupture	Arterial Rupture Detection Using LSFM ^2^	Macroscopic Findings of Arterial Rupture	Microscopic Findings of Arterial Rupture
1	95 Female	Subacute	Right parietal lobe	Completely detected	Small recess at the bifurcation of a minor branch	Partial rupture of the minor branch at bifurcation
2	94 Female	Acute	Right parietal lobe	Possible (Unclear visualization of distal rupture end)	Small recess	Complete rupture of the minor branch at bifurcation
3	64 Female	Acute	Left parietal lobe	Completely detected	Localized bleeding adhesion	Two complete rupturesof the minor branch at and near bifurcation
4	84 Female	Acute	Right temporal lobe	Completely detected	Normal	Partial rupture of the minor branch at bifurcation
5	92 Male	Acute	Right parietal lobe	Not detected (Out of imaging range)	Arachnoid defect consistent with burr hole scar	Partial rupture of the minor branch at bifurcation
6	91 Female	Encapsul-ated acute	Left parietal lobe	Not detected (Interference by bleeding)	Brain contusion with focal hematoma	Complete rupture inside focal hematoma

^1^ Subdural hematoma; ^2^ Light-sheet fluorescence microscopy.

## Data Availability

CUBIC imaging datasets generated and analyzed during the current study are available in the Zenodo, at https://doi.org/10.5281/zenodo.6575678 (accessed on 24 May 2022). Other data generated or analyzed during this study are included in this published article.

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
