# Peer review of "Detection and Morphological Analysis of Micro-Ruptured Cortical Arteries in Subdural Hematoma: Three-Dimensional Visualization Using the Tissue-Clearing Clear, Unobstructed, Brain/Body Imaging Cocktails and Computational Analysis Method"

_diagnostics, 2022, doi:10.3390/diagnostics12112875_

Round 1

Reviewer 1 Report

Excellent topic and interesting article

Author Response

Response to Reviewer 1 Comments

We deeply appreciate your positive peer review.

Reviewer 2 Report

This article presents a new method for the detection and morphological analysis of micro-ruptured cortical arteries in subdural hematoma (SDH), namely the tissue-clearing CUBIC (clear, unobstructed, brain/body Imaging cocktail and computational analysis) method combined with light-sheet fluorescence microscopy (LSFM) to reconstruct the two-dimensional and three-dimensional images. The CUBIC method was used in the autopsy of six cases of SDH with cortical artery rupture. Compared with conventional macroscopic and histological examination, the authors found that the CUBIC method provides several advantages in the investigation of micro-ruptured cortical arteries in SDH. Moreover, the authors discussed the limitations of the CUBIC method for the detection and morphological analysis of micro-ruptured cortical arteries in SDH, and they proposed the next research direction, using this new method to elucidate the pathogenesis of micro-ruptured cortical arteries in SDH.

This paper is innovative and the research design is appropriate. Further, the elucidation and discussion of the results are comprehensive. This paper provides a new method for the detection and analysis of SDH caused by cortical artery rupture, which can be used as an additional method for forensic autopsy, and also help to explore the pathogenesis of SDH.

This is a topic of interest to the researchers in the related areas but the paper needs some improvement before acceptance for publication. My detailed comments are as follows:

1. I think the Title does not fully summarize the main content of the article. Not only "Detection", but also "morphological analysis" should be included.

2. The sentence structure is too long. Such as, in page 2, Introduction, lines 63-67, the sentence should be modified into a few simple sentences.

3. In Figure 1, "Macroscopic imaging" also needs to be labeled in the flow chart.

4. In page 2, Materials and Methods 2. Please add a description of macroscopic imaging.

Therefore, our Overall Recommendation is: Accept after minor revision.

Author Response

Response to Reviewer 2 Comments

Thank you very much for your positive evaluation and helpful suggestions to our manuscript. We have incorporated the suggested changes into the manuscript to the best of our ability, as follows:

Point 1: I think the Title does not fully summarize the main content of the article. Not only "Detection", but also "morphological analysis" should be included.

Response 1: We completely agree with your statement. We changed the title to " Detection and Morphological Analysis of Micro-Ruptured…and Computational Analysis Method ".

Point 2: The sentence structure is too long. Such as, in page 2, Introduction, lines 63-67, the sentence should be modified into a few simple sentences.

Response 2: Including the part you indicated, We simplified the sentence in the following sections; Introduction (lines 65-68 in page 2), Materials and Methods (lines 121-122 and 124-127 in page 4)

Point 3: In Figure 1, "Macroscopic imaging" also needs to be labeled in the flow chart.

Response 3: We completely agree with your statement, so incorporated "Macroscopic imaging" into the flow chart.

Point 4: In page 2, Materials and Methods 2. Please add a description of macroscopic imaging.

Response 4: We completely agree with your statement, so added subsection " Macroscopic Imaging " (lines 85-92 in page 2).

Reviewer 3 Report

This manuscript investigates brain slabs of subdural hematoma victims by conventional imaging, histopathology, and the  "clear, unobstructed brain imaging cocktails and computational analysis" (CUBIC) method, developed by the same research group 8 years ago. The motivation of this study is well explained and the results are clearly presented. Apart from a few minor comments regarding Section 4, I can only express my gratitude for the opportunity to review such nice work.  

Suggestions for further improvements: 

1. Line 265: Please revise the sentence "Rupture of the cortical artery ... was suggested to be associated with the rupture occurrence [8,9,13]." Its message is unclear and the collective citation makes the reader's life even more difficult. I would provide a more explicit commentary, based on individually cited references. 

2. Line 280: It is not clear to me what is meant by "general conditions". 

3. Line 298: The term "fluorescent laser" is imprecise.

4. Lines 304-307: While discussing the limitations of CUBIC imaging, the authors suggest that the search for the arterial rupture site could rely on post-mortem CT angiography or intra-arterial fluid injection, thereby narrowing down the CUBIC field of view. The question arises, however, whether these techniques are gentle enough not to alter the rupture site. Please comment on this aspect based on ref. [52].

5. Lines 325-331: This paragraph seems to argue that the tissue-clearing protocol applied in this work is also suitable for tissue sample storage. Then, it raises the question of sample vulnerability, also touched earlier (lines 291-296), and does not end with a clear conclusion. 

Author Response

Response to Reviewer 3 Comments

Thank you very much for your positive evaluation and helpful suggestions to our manuscript. We have incorporated the suggested changes into the manuscript to the best of our ability, as follows:

Point 1: 1. Line 265: Please revise the sentence "Rupture of the cortical artery ... was suggested to be associated with the rupture occurrence [8,9,13]." Its message is unclear and the collective citation makes the reader's life even more difficult. I would provide a more explicit commentary, based on individually cited references. 

Response 1: We completely agree with your statement. We added sentences and rearranged citations for clarity (lines 273-279 in page 10).

Point 2: Line 280: It is not clear to me what is meant by "general conditions".

Response 2: We replaced "general conditions" with "common pathological conditions" (line 290 in page 10).

Point 3: Line 298: The term "fluorescent laser" is imprecise.

Response 3: Thank you for pointing this out. We corrected "fluorescence laser” (line 308 in page 11).

Point 4: Lines 304-307: While discussing the limitations of CUBIC imaging, the authors suggest that the search for the arterial rupture site could rely on post-mortem CT angiography or intra-arterial fluid injection, thereby narrowing down the CUBIC field of view. The question arises, however, whether these techniques are gentle enough not to alter the rupture site. Please comment on this aspect based on ref. [52].

Response 4: We completely agree with your statement. We added sentences about artificial rupture (lines 317-322 in page 11).

Point 5: Lines 325-331: This paragraph seems to argue that the tissue-clearing protocol applied in this work is also suitable for tissue sample storage. Then, it raises the question of sample vulnerability, also touched earlier (lines 291-296), and does not end with a clear conclusion.

Response 5: As you noted, this paragraph may be misleading. We are deleting “Our protocol is … sample storage.” because it is not the intent of this study to discuss the invaluable sample storage in this protocol. We have added a statement related to the brain fragility observed in this study(lines 342-343 in page 11).